# Use of volatile anesthetics for sedation in the ICU during the COVID-19 pandemic: A national survey in France (VOL'ICU 2 study)

Raiko Blondonnet[1,2]*, Aissatou Balde[1], Ruoyang Zhai[2], Bruno Pereira[3], Emmanuel Futier[1,2], Jean-Etienne Bazin[1], Thomas Godet[1], Jean-Michel Constantin[4], Céline Lambert[3], Matthieu Jabaudon[1,2,5]

1 Department of Perioperative Medicine, CHU Clermont-Ferrand, Clermont-Ferrand, France, 2 iGReD, CNRS, INSERM, Université Clermont Auvergne, Clermont-Ferrand, France, 3 Biostatistics Unit, DRCI, CHU Clermont-Ferrand, Clermont-Ferrand, France, 4 Department of Anesthesiology and Critical Care, Pitié-Salpêtrière Hospital, GRC 29, AP-HP, DMU DREAM, Sorbonne University, Paris, France, 5 Division of Allergy, Pulmonary, and Critical Care Medicine, Department of Medicine, Vanderbilt University Medical Center, Nashville, Tennessee, United States of America

* rblondonnet@chu-clermontferrand.fr

**Data Availability Statement:** All relevant data are within the paper and its Supporting Information files.

## Abstract

### Background

The COVID-19 pandemic has increased the number of patients in ICUs leading to a world-wide shortage of the intravenous sedative agents obligating physicians to find alternatives including inhaled sedation. Inhaled sedation in French ICU has been previously explored in 2019 (VOL'ICU study). This survey was designed to explore the use of inhaled sedation two years after our first survey and to evaluate how the COVID-19 pandemic has impacted the use of inhaled sedation.

### Methods

We designed a national survey, contacting medical directors of French ICUs between June and October 2021. Over a 50-item questionnaire, the survey covered the characteristics of the ICU, data on inhaled sedation, and practical aspects of inhaled ICU sedation for both COVID-19 and non-COVID-19 patients. Answers were compared with the previous survey, VOL'ICU.

### Results

Among the 405 ICUs contacted, 25% of the questionnaires were recorded. Most ICU directors (87%) knew about the use of inhaled ICU sedation and 63% of them have an inhaled sedation's device in their unit. The COVID-19 pandemic increased the use of inhaled sedation in French ICUs. The main reasons said by the respondent were "need for additional sedative" (62%), "shortage of intravenous sedatives" (38%) and "involved in a clinical trial" (30%). The main reasons for not using inhaled ICU sedation were "device not available" (76%), "lack of familiarity" (60%) and "no training for the teams" (58%). More than 70% of respondents were overall satisfied with the use of inhaled sedation. Almost 80% of

**Funding:** There was no funding for this work. This work was supported by internal funding of the Department of Perioperative medicine, CHU Clermont-Ferrand, France. Sedana Medical and Abbvie provided support in the form of fees for authors [MJ; JMC], but did not have any additional role in the study design, data collection and analysis, decision to publish, or preparation of the manuscript. The specific roles of these authors are articulated in the 'author contributions' section.

**Competing interests:** The authors have read the journal's policy and the authors of this manuscript have the following competing interests: MJ is a principal investigator of the SEvoflurane for Sedation in ARds (SESAR) (ClinicalTrials.gov Identifier: NCT04235608) and the ISCA study (ClinicalTrials.gov Identifier: NCT04383730), which are co-funded and funded, respectively, by grants from Sedana Medical. JMC and MJ received fees from Sedana Medical for participation in a scientific advisory panel; MJ received consulting fees from Abbvie. Neither Sedana Medical or Abbvie has no influence in the study and collection, analysis, and interpretation of data and in writing of the current study. Other authors have no competing interest. There are no patents, products in development or marketed products to declare. This does not alter our adherence to PLOS ONE policies on sharing data and materials.

**Abbreviations:** ARDS, Acute respiratory distress syndrome; COVID-19, 2019 Coronavirus disease; ECMO, Extracorporeal membrane oxygenation; ICU, Intensive care unit; NMBA, Neuromuscular blocking agent; STROBE, Strengthening the reporting of observational studies in epidemiology.

respondents stated that inhaled sedation was a seducing alternative to intravenous sedation for management of COVID-19 patients.

## Conclusion

The use of inhaled sedation in ICU has increased fastly in the last 2 years, and is frequently associated with a good satisfaction among the users. Even if the COVID-19 pandemic could have impacted the widespread use of inhaled sedation, it represents an alternative to intravenous sedation for more and more physicians.

## Introduction

Sedation is used daily in intensive care units (ICUs) to manage patients. It improves comfort and tolerance during mechanical ventilation, therapeutic interventions, or nursing care [1]. Sedation is usually performed with intravenous drugs such as propofol, midazolam or dexmedetomidine [2]. However, these products can cause serious side effects including delirium, propofol infusion syndrome and hemodynamic failure and increase the time to liberation from mechanical ventilation and the duration of stay in the ICU. All these effects are associated with increase in hospital morbidity and mortality. Therefore, clinical practice guidelines for analgesia and sedation in the ICU (e.g., the Pain, Agitation/sedation, Delirium, Immobility and Sleep disruption (PADIS) guidelines [1]) have consistently focused on early rehabilitation and rapid ventilator liberation and have suggested the use of non-benzodiazepines drugs even if adaptation of ventilator settings should be systematically considered before administering additional medications [2]. The ideal sedative drug should be effective with few adverse effects, low accumulation, and a quick awakening at the end of administration.

Since the beginning of 2020, an unprecedented pandemic changed the daily management of ICUs worldwide: coronavirus disease 2019 (COVID-19) [3]. This pulmonary infection is caused by the severe acute respiratory syndrome coronavirus (SARS-CoV) -2, and has suddenly increased the number of patients hospitalized in ICUs due to respiratory failure. In France, almost 100.000 individuals were admitted to ICUs [4] and most of them needed intubation to support respiratory function. The pandemic posed a major challenge to health-care systems because of the need for intensive care therapy and mechanical ventilation including sedation. Because of this unusual situation, intensivists had to use more sedative products leading to a worldwide shortage in critical supplies such as main intravenous sedative agents [5–9]. Consequently, physicians had to find alternatives to intravenous sedation including inhaled sedation [10]. Inhaled sedation is performed with volatile agents such as sevoflurane, desflurane or isoflurane. Inhaled volatile agents are an abundant resource and an easily implementable solution for providing ICU sedation [11]. The recent development of anesthetic reflectors, such as the *Anaesthetic Conserving Device* (*Sedaconda-ACD*, Sedana Medical, Danderyd, Sweden) and the *Mirus* (Carelide GmbH, Mouvaux, France), has allowed delivering inhaled sedation in the ICU [12–14]. These two devices are inline miniature vaporisers with humidification and antiviral filter properties. For the *Anaesthetic Conserving Device*, the titration of the desired sedation level is performed manually whereas the *Mirus* system adjusts infusion rates to deliver volatile anesthetics through the automatic control of anesthetic concentration targets [15]. Indeed, in some national guidelines including Germany's, the use of volatile agents in the ICU is an option [16]. Furthermore, using volatile anesthetics could now be considered for specific acute respiratory distress syndrome (ARDS) patients to reduce

emergent delirium and cumulative propofol doses [2]. Recently, isoflurane received national approval from the French Agency for the Safety of Health Products (ANSM, for *Agence nationale de sécurité du médicament et des produits de santé*), among other European agencies, for inhaled sedation in the ICU.

A previous study from our group explored inhaled sedation practices in French ICUs in 2019 and demonstrated that, even if most physicians were familiar with inhaled sedation, it was underused because of a lack of available devices, physicians knowledge, and supporting literature [17]. Since March 2020, few studies have cared about the potential benefits of inhaled sedation in COVID-19 patients. Nevertheless, extant data suggest that inhaled sedation could be used safely in these patients [10, 11, 18]. Indeed, volatile agents may also provide important pulmonary benefits for COVID-19 patients with ARDS that could improve gas exchange, and reduce the time spent on a ventilator [11]. Currently, recent studies have shown that inhaled sedation in COVID-19 patients reduces the need for both intravenous sedation and opïods [18–20]. Furthermore, several clinical trials are ongoing to study the use of inhaled sedation in ICU patients with ARDS secondary or not to COVID-19 [21].

Therefore, this survey was designed to explore and describe the use of inhaled sedation two years after our first survey about inhaled sedation in France [17], and to evaluate how the COVID-19 pandemic has impacted the use of inhaled sedation by French physicians working in ICUs.

## Methods

### Survey development

This investigator-initiated survey was approved by an independent Ethics Committee (CERAR IRB 00010254–2021–128). A 50-item questionnaire was developed with questions designed by the authors (RB, AB, MJ) (**S1 File**); The survey covered four categories: general characteristics of the ICU, general data of COVID-19 patients in ICU, general data on inhaled sedation, and the practical aspects of inhaled ICU sedation for both COVID-19 and non-COVID-19 patients. Some answers were compared with the previous survey, VOL'ICU [17].

### Survey sample

All French ICUs were identified and contacted. Pediatric ICUs were excluded from the survey. All the same ICUs were contacted in this study as were contacted in the VOL'ICU study in order to compare the answers between the two studies [17]. The current survey was conducted between June and October 2021. After short information about the survey design and objectives, the medical director of each ICU was questioned exclusively. Completion of the survey took approximately ten minutes. The first contact with the ICU directors to complete the survey was by email. In the case of non-response, a second attempt at contact was made by email or phone with the ICU director in order to recorded the survey and followed by a last call when necessary.

### Statistical analysis

Statistical analysis was performed in compliance with the Strengthening the Reporting of Observational Studies in Epidemiology (STROBE) checklist [22]. Statistical analysis was performed using Stata software (version 15; StataCorp, College Station, Texas, USA). All tests were two-sided, with a Type I error set at 0.05. Categorical data were expressed as frequencies and associated percentages, and quantitative data as mean ± standard deviation or median [1st quartile; 3rd quartile], according to statistical distribution. The results of the two surveys were

compared using the chi-squared test or Fisher's exact test (qualitative variables only). The paired qualitative data were compared by the Stuart-Maxwell test.

## Results

### General characteristics

Among the 405 French ICUs [23], 31 pediatric ICUs were excluded and 374 adult ICU directors were questioned. A total of 25% (102/405) of the questionnaires were recorded, 95% (97/102) electronically and 5% (5/102) orally. The general characteristics of ICU respondents and geographical distribution of respondents are reported in **S2 and S3 Files**. Of the answers, 52% (53/102) came from general hospitals, 38% (39/102) from teaching hospitals, 9% (9/102) from private medical centers and 1% (1/102) from military hospitals. Among the participating ICUs, 89% (91/102) were mixed (medical and surgical) ICUs, 10% (10/102) were only medical ICUs and 1% (1/102) was a burn center. The majority of the ICU respondents managed COVID-19 patients (95%, 97/102).

### Practical aspects of inhaled ICU sedation

Among the respondents, 87% (89/102) stated that they knew about the use of inhaled sedation in the ICU. Among the 89 respondents, 9% (8/89) discovered inhaled sedation during the COVID-19 pandemic. Of these, 91% (81/89) knew about the *SedaConda-ACD* device and 10% (9/89) knew the *Mirus* device. Sixty-three percent (56/89) of respondents who knew about the use of inhaled sedation in the ICU reported disposing of a specific device in their unit. Among these respondents, 84% (47/56) declared they performed inhaled sedation with *Sedaconda-ACD*, and 14% (8/56) used *Mirus*. Five percent of respondents (3/56) declared to own both. Thirty-four percent (19/56) of respondents answered that they had acquired a device since the pandemic. If neither the *Sedaconda-ACD* nor the *Mirus* were available in their hospital, 11% (6/56) of respondents declared that they borrowed anesthesia ventilators from the operating room to deliver volatile anesthetics in the ICU.

According to all respondents, 35% (36/102) reported performing inhaled sedation in the ICU before the beginning of the COVID-19 pandemic and 47% (48/102) of respondents reported using inhaled sedation in the ICU since the COVID-19 pandemic began. Twenty-five percent of the respondents (25/102) declared that they had modified their use of inhaled sedation during COVID-19. The two main reasons for changing the use of inhaled sedation were: a more frequent use independent of COVID-19 and a more frequent use with ARDS patients. Among the respondents who did not use inhaled sedation, 34% (20/58) said they were favorable to developing inhaled sedation in their unit within the next two years.

During the COVID-19 pandemic, the reasons declared by the respondents to use inhaled sedation were in descending order: the need for additional sedative (62%, (28/45)), a shortage of intravenous sedatives (38%, (17/45)),the unit was involved in a clinical trial about inhaled sedation (30% (17/45)) and interest in the system (27%, (12/45)).

### Indications and contraindications for inhaled ICU sedation

Twenty-seven percent of respondents (15/56) declared that inhaled sedation was performed by all physicians working in their units. The majority of the respondents answered they used inhaled sedation in the ICU in less than 20 patients per year before and after the beginning of COVID-19 (**Fig 1**).

Main reasons for not using inhaled ICU sedation were as follows: "device not available" (76%, (47/62)) and "lack of familiarity about the technique" (60%, (37/62)) and "lack of

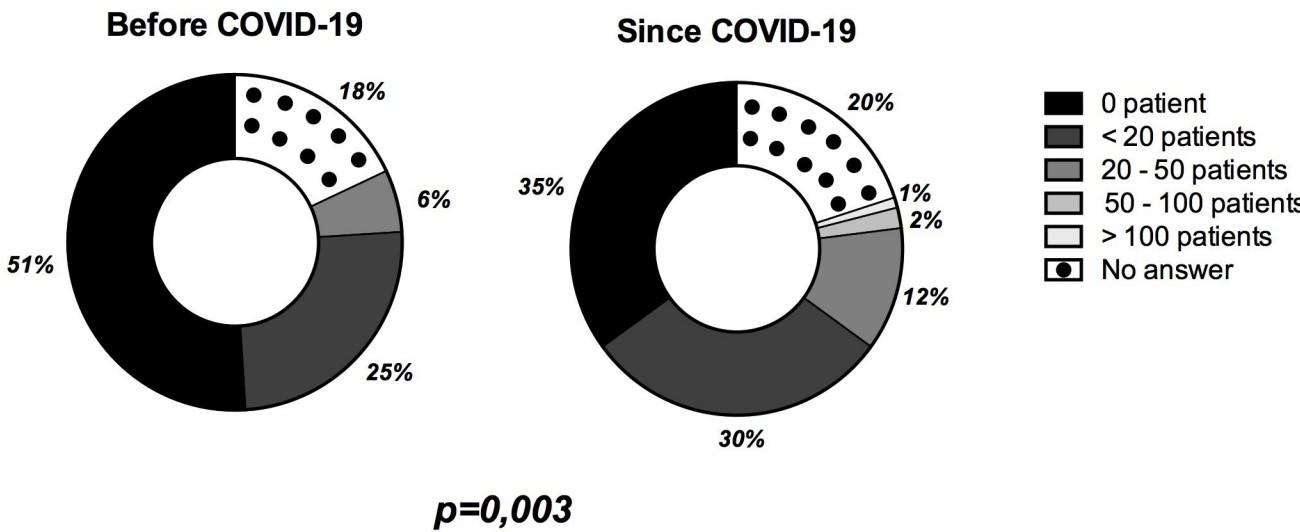

**Fig 1. Number of patients per year sedated with volatile sedation in ICU declared by the respondents (n = 102).** Data are represented in numbers of ICU respondents.

formation for both the physicians and the nurses" (58%, (36/62) each) and "halogenated-induced atmospheric pollution" (27%, (17/62)) (**Fig 2**).

About indications, the main reported by respondents were: "failure of intravenous sedation" (74%, (39/53)), "severe asthma" (64%, (34/53)), and ARDS (49%, (26/53)). The other indications mentioned are summarized in **Fig 3**. The main advantages for using volatile sedation answered by respondents were: bronchodilation (83%, (44/53)) and usability (64%, (34/53)).

Absolute contraindications for inhaled sedation were reported by 94% (68/72) of the respondents (**Fig 4**). Thirty-nine percent (22/57) of the respondents declared that they had already at least one adverse effect attributable to volatile anesthetics, mainly reporting diabetes insipidus, malignant hyperthermia, and hypercapnic acidosis.

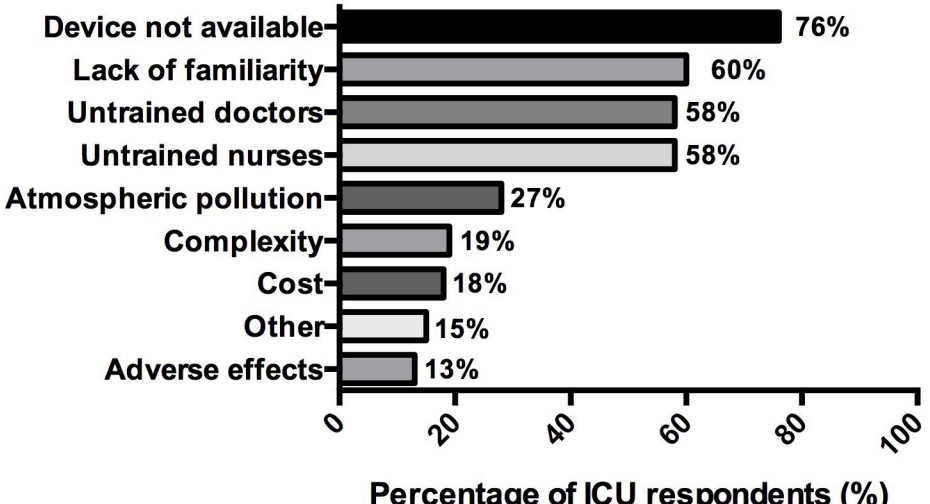

**Fig 2. Reasons declared by the respondents for not using inhaled sedation in ICU (n = 62).** Data are represented in %.

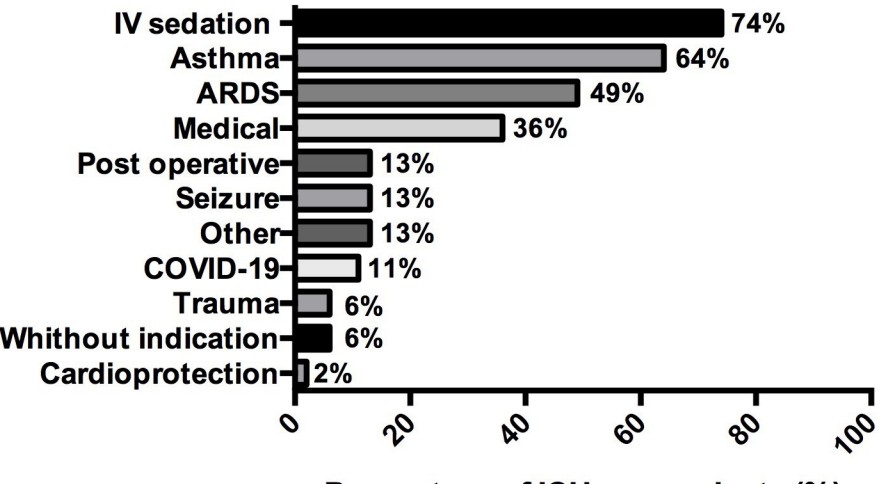

**Fig 3. Indications declared by the respondents for using inhaled sedation in ICU (n = 53).** Data are represented in %.

## Inhaled sedation in practice

Eighty-two percent of respondents (83/101) declared that they had a written protocol for sedation in their institution whereas 64% (34/53) of units had a specific protocol for inhaled sedation.

Among units using inhaled sedation, 41% (19/46) of the respondents said they received specific training on inhaled sedation, either from the companies developing the devices in 84% (16/19) of the respondents or through scientific conferences in 21% (4/19) of the respondents.

Sevoflurane and isoflurane were the main drugs used, as reported by the respondents (94%, (30/32) and 22%, (7/32) respectively) and 3 respondents (10% (3/32)) answered they used both. Four respondents (13%, (4/32)) said they used desflurane. To manage sedation,

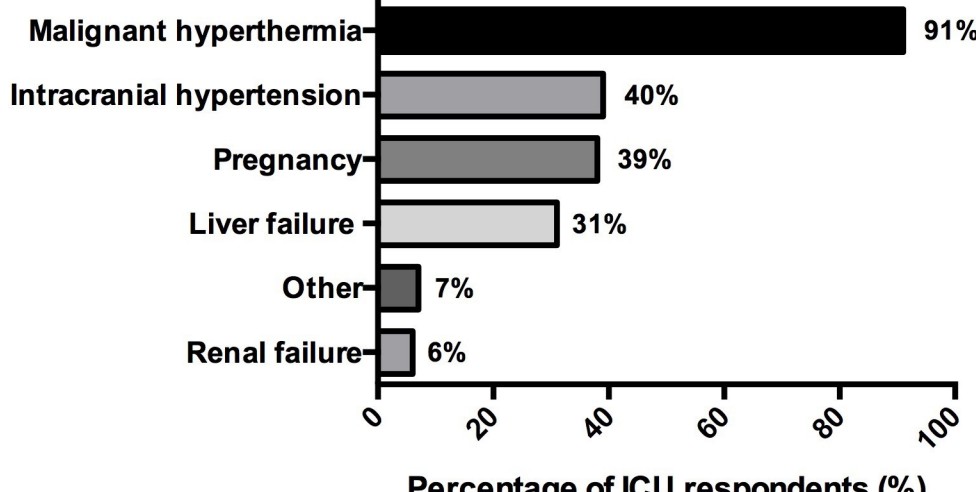

**Fig 4. Contraindications declared by the respondents for using inhaled sedation in ICU (n = 68).** Data are represented in %.

respondents reported achieving a target exhaled gas fraction with sevoflurane of 1.0 [0.8–1.2] (35/56) and 0.5 [0.4–0.7] (27/56) for deep sedation and light sedation, respectively. All respondents said that they used inhaled sedation with controlled ventilation mode. Forty-six percent of them (25/54) stated that they also used inhaled sedation during pressure support ventilation in intubated patients. Two respondents (4% (2/54)) declared that they had already used inhaled sedation during non-invasive ventilation.

All the respondents reported that they usually combined opioid-based analgesia with inhaled sedation, 65% (35/54) with sufentanil, 37% (20/54) with remifentanil and 6% (3/54) with fentanyl or morphine. One-third of them (18/54) answered that they combined gas with continuous intravenous sedative hypnotic, such as propofol (94%, (17/18)), midazolam (61%, (11/18)), ketamine (33%, (6/18)) or dexmedetomidine (17%, (3/18)).

When asked how they measured sedation objectives, most of the respondents (93%, (77/83)) answered using validated sedation scales or scores (such as The Richmond Agitation Sedation Scale (RASS)) rather than end-tidal gas concentration monitoring (42%, (35/83)). Forty-eight percent of the respondents (40/83) used the bispectral index (BIS-ASPECT-A-2000; Aspect Medical Systems, Norwood, USA). Only one ICU director reported that he measured plasma concentrations of volatile anesthetics or their metabolites when monitoring inhaled sedation.

According to the ICU directors' answers, 65% (35/54) interrupted inhaled sedation when patients began the process of weaning from ventilation and 37% (20/54) did not specifically set any maximal duration for inhaled sedation in their ICU patients.

Overall satisfaction with the use of inhaled sedation among users is represented in **Fig 5**. Seventy-nine percent (69/87) of the respondents declared that inhaled sedation could be an

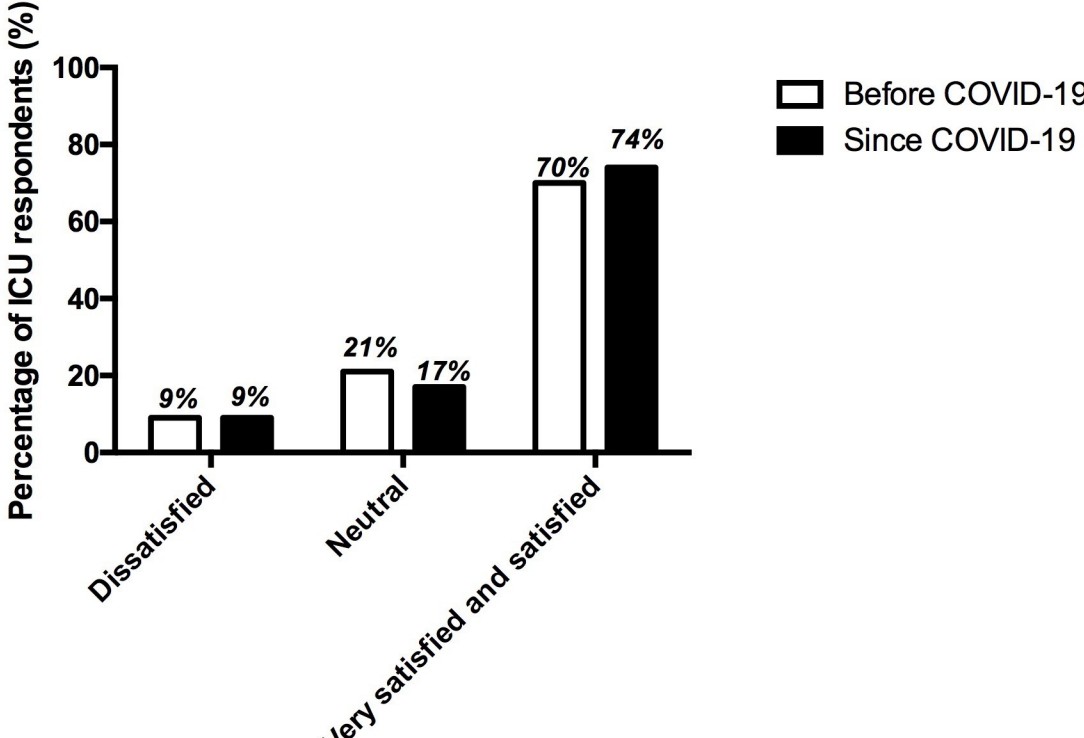

**Fig 5. Overall satisfaction of respondents regarding the use of inhaled sedation before (n = 43) and since the COVID-19 (n = 53).** Data are represented in %.

interesting alternative to intravenous sedation and especially for COVID-19 patients for 76% (62/82) of the respondents.

## Evolution of inhaled sedation use compared to VOL'ICU study

The comparison between the data from VOL'ICU and VOL'ICU2 is summarized in **Table 1**. Since the beginning of COVID-19, the respondents declared that they know more about inhaled sedation in the ICU and dispose more frequently of use at least one of the devices for delivering inhaled sedation. The indications for the use of inhaled sedation declared by the respondents were unchanged between VOL'ICU and VOL'ICU2 except for an increased use for medical indications since the COVID-19 pandemic. The two main volatile anesthetics used remained sevoflurane and isoflurane, but 13% of respondents declared to use desflurane in VOL'ICU2 compared to zero respondents in VOL'ICU. No significant modification of the rate of adverse effects was shown between the two studies.

## Discussion

This survey is the first to re-evaluate the use of inhaled sedation after the COVID-19 pandemic in French ICUs. This survey shows that, in 2021, both knowledge of and interest in inhaled sedation are growing significantly among French intensivists. Nonetheless, the lack of devices with which to perform inhaled sedation in units and the insufficient training of both medical and paramedical teams remain barriers to widespread use of inhaled sedation.

The COVID-19 pandemic has led to a worldwide shortage of intravenous sedative drugs, prompting intensivists to search for alternatives to sedate ICU patients [5–9]. Interestingly, French intensivists mainly justified the use of inhaled sedation in their units during COVID-19 with the need for additional sedatives [10]. Thus, COVID-19 patients, especially when they develop ARDS, need higher doses of sedatives to reach the sedation objective compared to non-COVID-19 patients [24]. This element may also explain the increase of use of inhaled sedation with or without another hypnotic in our survey. The use of inhaled sedation due to shortages was declared by only one-third of the respondents as well as the involvement of the unit in clinical trials on inhaled sedation. This point suggests that the popularity of inhaled sedation may be increased because of its intrinsic sedative characteristics. The respondents also reported potential interest in the bronchodilator effect and the manageability of inhaled sedation which could explain why volatile anesthetics may be a more popular option in the management of COVID-19 patients. Indeed, the efficacy of volatile anesthetic is also demonstrated in patients with refractory life-threatening status asthmaticus [25]. Furthermore, some studies also suggest that volatile anesthetics has a good manageability, avoids tachyphylaxis and perhaps anti-inflammatory effects [26, 27]. For all these reasons, volatile sedation may become, theoretically, an ideal option for intensivists in order to manage sedation. Furthermore, a large number of clinical trials studying the effects of inhaled sedation with either sevoflurane or isoflurane on major clinical outcomes in both non-COVID-19 and COVID-19 patients are actually enrolling patients (ClinicalTrials.gov: NCT04235608, ClinicalTrials.gov: NCT04415060, ClinicalTrials.gov: NCT04341350). In addition to providing important results about inhaled sedation in ICU, these trials may contribute to help participating centers to be more familiar with this technique.

However, we noted that scarcity was not the main reason for the increased use of volatile sedation in ICUs since the pandemic started: indeed, respondents reported that they used volatile anesthetics to achieve better sedation and to manage patients with ARDS. Volatile anesthetics contraindications in ICU seem known by almost all participants even if answers varied. Malignant hyperthermia was the contraindication mainly responded by the physicians and

**Table 1. Comparisons between the answers provided by the respondents in the VOL'ICU (n = 187) and VOL'ICU2 (n = 102) studies.** Results are presented as numbers (with associated percentages). Percentages were rounded to the nearest whole number depending on whether the value after the decimal was greater than or less than 5.

| VARIABLE | VOL'ICU | VOL'ICU 2 | p value |
|---|---|---|---|
| **Knowledge of inhaled sedation, n (%)** | | | |
| Yes | 137 (73) | 89 (87) | 0.006 |
| No | 50 (27) | 13 (22) | |
| **Availability of the device in the unit, n (%)** | | | |
| Yes | 40 (21) | 56 (55) | < 0.001 |
| No | 147 (79) | 45 (55) | |
| **Indications, n (%)** | | | |
| Medical | 13 | 36 | 0.01 |
| Post-operative | 13 | 13 | 0.92 |
| ARDS | 65 | 49 | 0.13 |
| Trauma | 3 | 6 | 0.63 |
| Failed sedation | 75 | 74 | 0.88 |
| Asthma | 75 | 64 | 0.26 |
| Status epilepticus | 10 | 13 | 0.64 |
| Cardioprotection | 13 | 2 | 0.08 |
| No specific indication | 3 | 6 | 0.63 |
| **Reasons declared for not using inhaled sedation, n (%)** | | | |
| No device available | 39 | 76 | < 0.001 |
| Lack of medical training | 22 | 58 | < 0.001 |
| Lack of paramedic training | 16 | 58 | < 0.001 |
| Complexity | 12 | 19 | 0.14 |
| Ecological concern | 7 | 27 | < 0.001 |
| Adverse effects | 1 | 13 | < 0.001 |
| Lack of familiarity | 35 | 60 | < 0.001 |
| Cost | 21 | 18 | 0.58 |
| **Sedation protocol, n (%)** | | | |
| *Intravenous sedation* | | | |
| Yes | 157 (84) | 83 (82) | 0.7 |
| No | 30 (16) | 18 (18) | |
| *Inhaled sedation* | | | |
| Yes | 17 (45) | 34 (34) | 0.23 |
| No | 21 (55) | 67 (66) | |
| **Volatile anesthetics, n (%)** | | | |
| Sevoflurane | 88 | 94 | 0.45 |
| Isoflurane | 20 | 22 | 0.85 |
| Desflurane | 0 | 13 | 0.04 |
| **Adverse effects, n (%)** | | | |
| Yes | 11 (28) | 22 (39) | 0.26 |
| No | 29 (73) | 35 (61) | |

clearly documented by studies in the literature [28]. The other contraindications remained unclear for ICU's practitioners and it could be explained by a very scarce literature about at-risk patient categories. Inhaled sedation should be avoided when there is a risk of increased intracranial pressure. However, inhaled sedation could be used, in some circumstances, both in neurosurgical and neurocritical patients [29]. Indeed, volatile anesthetics are known for

their antiepileptic properties and may have therapeutic benefits in patients with refractory status epilepticus [15, 30–32]. Furthermore, volatile anesthetics have also been studied in patients with stroke and subarachnoid hemorrhage, but with less promising results [31]. During pregnancy, the use of sevoflurane seems safe, but the "precautionary principle" should be applied to limit exposure as much as possible [33]. Interestingly, the number of adverse effects reported by the users of inhaled sedation did not statistically change between VOL'ICU and VOL'ICU2. Some respondents reported suspected or confirmed cases of diabetes insipidus, but also malignant hyperthermia and hypercapnia with acidosis. Unfortunately, the literature about adverse events related to inhaled sedation use is limited and the most often, the causality remains under debate. Nevertheless, the prolonged use of inhaled sedation (e.g., for >48 h) has shown good safety with equivalent effects on hemodynamic stability, no hepatorenal toxicity, and possibly less agitation compared to intravenous agents [34–37]. Even if large-scale studies are urgently needed to confirm safety of inhaled ICU sedation, the multicenter randomized controlled SEDACONDA study found that ICU sedation with isoflurane for up to 54 h is safe and efficacious as a sole sedative and non-inferior to propofol in maintaining targeted sedation levels [38].

Managing inhaled sedation could be frightening for ICU teams when starting, notably with regards to the sedation levels targeted. Interestingly, most of the respondents mainly managed inhaled sedation in ICU patients using validated sedation scales or scores (e.g., RASS) rather than end-tidal gas concentration monitoring. Anyway, monitoring sedation level with a validated tool, titrating all sedative agents and reassessing the target sedation level several times a day are needed to manage sedation in ICU whatever the sedatives [2]. Meanwhile, modern approaches of "analgesia-first" or "analgesic-based sedation" are growing favoring the use of an analgesic before a sedative for pain management in ICU [2]. As intravenous sedation has to be titrated by ICU teams, inhaled sedation should be managed by an "inhaled titration" of volatile anesthetics agents using scales such as the RASS. All the more, the end-tidal gas concentration monitoring and RASS are correlated in ICU patients [39]. No data are available to date on the safety and efficacy to manage inhaled sedation only using clinical sedation scores. In some specific patients, such as ARDS patients with neuromuscular blockade (NMBA), management of inhaled sedation should probably integrate end-tidal gas concentration monitoring and/or instruments. Indeed, among patients receiving NMBAs, neither the gold standard for pain assessment (i.e., the patient's self-report) or recommended behavioral measures can be used. Alternative approaches for pain and sedation assessment in paralyzed patients are being explored, such as the analgesia nociception index or the pupillary pain index [2]. Unfortunately, we did not investigate in our survey if the scales or scores were evaluated by nurses or by doctors. Indeed, the use of nurse-directed analgesia/sedation protocols, which enable bedside nurses to adjust opioids and sedatives can reduce drug exposure and shorten weaning from mechanical ventilation and ICU discharge.

The major restraining factors for a more widespread use of inhaled sedation remains the availability of the device and the training of both medical and paramedical teams. Nevertheless, devices to perform inhaled sedation are available in 36% of additional ICU and 10% of additional ICU just started using inhaled sedation, compared to the results of our first survey. The same restraining factors were answered by the respondents in 2019 in the VOL'ICU study. These points strengthen the importance of education programs of the caregivers who work in ICUs to decrease the fear of using a "new" sedation technique whatever their specialty track. Indeed, some other european and non-european intensivists with different specialty track trains compared to France use inhaled sedation in ICU [37, 38]. These elements should encourage the practitioners to reinforce the training and pedagogical requirements about inhaled sedation in their units and to integrate inhaled

sedation into multifaceted bundles of sedation in the ICU [2]. The education of both prac-
titioners and nurses is crucial to developing the use of inhaled in the ICU. Hopefully, the
use of volatile anesthetics can now be considered for specific ARDS patients to reduce
emergent delirium and cumulative propofol doses [2]. Furthermore, isoflurane received
approvals from 15 European medicines agencies for inhaled sedation in ICU which will
result in an increased use of inhaled sedation. This increase in the use of inhaled sedation
should be supported by the writing of a specific protocol for inhaled sedation which is
insufficiently present in the units to date. Indeed, oversedation remains common in many
ICUs such that an sedation protocol is frequently beneficial for patients. Another point is
that even if industries provided many devices all around France during the COVID-19
pandemic, some ICUs could have experienced shortages in some devices or consumables,
thus restraining inhaled sedation use and explaining that 47/62 respondents did not use
the device due to lack of availability. Furthermore, due to the difference in price between
the *Sedaconda-ACD* and *MIRUS* devices, the *MIRUS* may not be suitable for short-term
acquisition and could be an explanation for the clear dominance of the *Sedaconda-ACD*
use in our study.

Extracorporeal membrane oxygenation (ECMO) is used for patients with severe respira-
tory failure and received particular attention during the COVID-19 pandemic [40]. Usually,
patients are sedated using intravenous sedative drugs to reduce oxygen consumption. How-
ever, patients undergoing ECMO require more intravenous sedative drugs because of the
loss of these drugs via the ECMO circuit [41, 42]. During the shortage in intravenous seda-
tives, the sedation of ECMO patients could be challenging. Use of inhaled sedation in ARDS
patients treated with ECMO is relatively novel and raises several feasibility and safety ques-
tions. Nevertheless, volatile anesthetics administration was effective finor ARDS patients
undergoing ECMO [43–45]. Volatile anesthetics could be delivered through the mechanical
ventilation [43] or the ECMO [44] circuits. Inhaled sedation could be used for patients
undergoing cardiac surgery with administration of volatile anesthetics through the *Seda-
conda*-ACD connected directly to the extra corporeal circulation [46]. However, more data
on the use of this method and further clinical studies are needed before such a method can
be generalized.

The risk of room and environmental pollution remains a limiting factor among the respon-
dents for the use of inhaled sedation in ICU. It is true that average threshold limit concentra-
tions for volatile anesthetics differ significantly between countries or are not even defined at
all, leading to raising concerns among teams who work in ICU [47]. Actually multiple studies
reported room pollution far below the recommended exposure limits even in countries with
lowest recommendations [33, 48, 49]. Concerns are often historically founded, when personnel
were exposed to high concentrations of evidently toxic substances, which were used in rooms
without air-conditionings and gas scavenging systems. However, based on currently available
data, there is no significant pollution when the anesthetic reflectors are correctly set up and
used in accordance with recommendations from their manufacturers. Besides, in order to
decrease room pollution, the Swedish and American authorities recommend that ICU rooms
are equipped with air conditioning that has at least 6 air changes per hour. Indeed effective air
conditioning is an effective system to maintain low values of waste volatile anesthetics below
the recommendations [33, 49]. Additionally, devices such as the *Sedaconda-ACD* reflects mois-
ture back to the patient, but also reflects up to 90% of the volatile anesthetics by adsorbing and
releasing the volatile using a proprietary carbon filament reflecting medium. This reflection
reduces the total amount of volatile anesthetics needed, reducing that which is exhausted or
scavenged [50]. Furthermore, activated carbon systems connected to the expiratory branch of
the ventilator or active scavengers connected to the vacuum system exist worldwide to capture

the volatile anesthetics no longer absorbed by the devices; ideally, these residuals could be recycled in the future [47, 51–53].

Our survey has several limitations. First, with 102 survey responses, the participation rate in this second survey was lower than in our previous survey (n = 187, 50% response rate). With the second survey, we were only able to reach about 27% of potentially eligible adult ICUs in France [16]. Since the beginning of the COVID-19 crisis, the French healthcare system has been challenged with reorganizations of critical care all around the country including major physician and nurse turnovers [54]. Furthermore, a lower interest in this survey could be explained by the amount of work for some ICU directors due to a number of COVID-19 cases that remain high in some regions. Nevertheless, the characteristics of the answering ICUs were comparable to those of our previous survey. Furthermore, we used the same information channels to distribute the survey link and the time-period (summer) was identical to our first survey. Second, declarative surveys can only provide limited information due to intrinsic bias which could be avoided with an observational study. Nevertheless, the validity of our findings is highlighted by the design of the study in which only ICU directors were questioned to limit both the non-response rate and the response bias. Finally these findings, which reflect the use of inhaled sedation as reported by French intensivists, may not be extrapolated to other countries with distinct ICU organizations.

## Conclusion

In conclusion, our study shows that the use of inhaled sedation in ICU has increased since 2019, and is frequently associated with a good satisfaction among the users. Especially since the COVID-19 pandemic, the use of inhaled sedation could represent an alternative to intravenous sedation for more and more French physicians. Nonetheless, both lack of devices available in the units and insufficient training of ICU teams remain the two major restraining factors for the use of inhaled ICU sedation.

## Supporting information

**S1 Data.**
(XLSX)

**S1 File. Survey questionnaire.**
(DOCX)

**S2 File. Characteristics of the centers where respondents worked and characteristics of the non-COVID-19 and COVID-19 patients estimated by the respondents.**
(DOCX)

**S3 File. Geographical distribution and epidemiological data on respondents.**
(XLSX)

## Acknowledgments

The authors would like to thank all the respondents of the survey.

## Author Contributions

**Conceptualization:** Raiko Blondonnet, Aissatou Balde, Jean-Michel Constantin, Matthieu Jabaudon.

**Data curation:** Raiko Blondonnet, Matthieu Jabaudon.

**Formal analysis:** Bruno Pereira, Céline Lambert.

**Funding acquisition:** Raiko Blondonnet, Matthieu Jabaudon.

**Investigation:** Raiko Blondonnet, Aissatou Balde, Jean-Michel Constantin, Matthieu Jabaudon.

**Methodology:** Raiko Blondonnet, Aissatou Balde, Bruno Pereira, Jean-Michel Constantin, Céline Lambert, Matthieu Jabaudon.

**Project administration:** Raiko Blondonnet, Aissatou Balde, Matthieu Jabaudon.

**Supervision:** Raiko Blondonnet, Aissatou Balde, Matthieu Jabaudon.

**Validation:** Raiko Blondonnet, Jean-Michel Constantin, Matthieu Jabaudon.

**Writing – original draft:** Raiko Blondonnet, Aissatou Balde, Céline Lambert, Matthieu Jabaudon.

**Writing – review & editing:** Raiko Blondonnet, Aissatou Balde, Ruoyang Zhai, Bruno Pereira, Emmanuel Futier, Jean-Etienne Bazin, Thomas Godet, Jean-Michel Constantin, Céline Lambert, Matthieu Jabaudon.

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
