## [Decision Letter · Decision Letter 0]

15 Sep 2022

PONE-D-22-06434Use of volatile anesthetics for sedation in the ICU after the COVID-19 pandemic: A national survey in France (VOL’ICU 2 study)PLOS ONE

Dear Dr. Blondonnet,

Thank you for submitting your manuscript to PLOS ONE. After careful consideration, we feel that it has merit but does not fully meet PLOS ONE’s publication criteria as it currently stands. Therefore, we invite you to submit a revised version of the manuscript that addresses the points raised during the review process.

ACADEMIC EDITOR:

Please carefully review your references.

We look forward to receiving your revised manuscript.

Kind regards,

Silvia Fiorelli

Academic Editor

PLOS ONE

Journal Requirements:

“The authors have read the journal’s policy and the authors of this manuscript have the following competing interests: MJ is a principal investigator of the SEvoflurane for Sedation in ARds (SESAR) (ClinicalTrials.gov Identifier: NCT04235608) and the ISCA study (ClinicalTrials.gov Identifier: NCT04383730), which are co-funded and funded, respectively, by grants from Sedana Medical. JMC and MJ received fees from Sedana Medical for participation in a scientific advisory panel; MJ received consulting fees from Abbvie. Other authors have no competing interest. There are no patents, products in development or marketed products to declare”

We note that one or more of the authors are employed by a commercial company: SESAR, Sedana Medical

Reviewers' comments:

Reviewer's Responses to Questions

**Comments to the Author**

1. Is the manuscript technically sound, and do the data support the conclusions?

Reviewer #1: Yes

Reviewer #2: Partly

2. Has the statistical analysis been performed appropriately and rigorously? 

Reviewer #1: Yes

Reviewer #2: N/A

3. Have the authors made all data underlying the findings in their manuscript fully available?

Reviewer #1: Yes

Reviewer #2: Yes

4. Is the manuscript presented in an intelligible fashion and written in standard English?

Reviewer #1: Yes

Reviewer #2: Yes

5. Review Comments to the Author

Reviewer #1: 1. Title should be changed - from "...in the ICU AFTER COVID 19..." in "...in the ICU DURING COVID 19...." because the survey questions were related to COVID19 period.

2. Figure no. 1 has to changed into a more clear one. Legend should be chnaged (grey shades used can lead to mistakes - "no answer" and "> 100 patients" having a similar colour). Also the legend is not intelligible and not totally compatible with the text description.

Reviewer #2: I would like to express my gratitude for the opportunity to review the submitted intriguing work on the use of volatile sedation in France. Unfortunately, the work in its current form still has a number of shortcomings.

I am sorry, but I have to criticise the inadequate referencing work. The study by your French colleague (reference 5 ) does not describe an actual shortage of sedatives! The colleagues describe aptly in their introduction to your study project: “Another major concern was to ensure access to mandatory anesthetics drugs medications, or neuromuscular blockade agents. In response to such shortages, we decided to diversify our sedative agents panel and thus to use volatile agents…”. However, authors should not be guided by the title of a paper without allowing its content to be presented in the referenced study! So far, the manuscript fails to provide appropriate references for the presentation of a real drug shortage. This applies equally to the corresponding section of the discussion.

How do the authors conclude that volatile substances are an "abundant" resource? Whereas, according to them, intravenous sedatives are a scarce resource?

I'm not sure, but if I'm not mistaken, the MIRUS is the MIRUS device from the company Technologie Institut Medizin, which is distributed in by carelide, Dahlhausen, Anandic or Pall?

In another passage, the authors cite a well-done Narrative Review by the esteemed colleague Jerath on Volatile Sedation in COVID-19 (reference 11), which raises the hypothesis outlined, but does not substantiate it with appropriate COVID-19-specific data. Such data should present masterful performance 3 months after the global COVID-19 outbreak as well. A corresponding referencing with current data would be exciting.

Regarding the acquisition of medical devices, it seems that 34% new acquisitions during the pandemic is a quite high rate of observation. However, it should be discussed that the purchase of a Sedaconda ACD membrane is around 16€ and a MIRUS device costs several thousand euros. Therefore, the MIRUS Device is not suitable for short-term acquisitions and would be an explanation for the clear dominance of the Anaesthetic conserving Device: AnaConDa/SeDaConDa.

How do the authors explain that 47/62 respondents did not use the device due to lack of availability, when the majority of them used Sedaconda as a patient-specific single-use material?

The authors disclose early on that numerous uses of volatile sedatives are under study conditions. Has it been evaluated on your part whether the center determining intravenous gas concentrations of volatiles is a study condition determination? I can't explain it any other way. And it does seem a bit confusing to someone not versed in the field.

“Thus, COVID-19 patients, especially when they develop ARDS, need higher doses of sedatives to reach the sedation objective compared to non-COVID19 patients [11].” As noted above, this source does not reflect the aggravated sedation per se, here the evidence has already been provided elsewhere: https://doi.org/10.1371/journal.pone.0253778/ https://doi.org/10.1371/journal.pone.0253778.

„The respondents also reported potential interest in the bronchodilator effect and…” also the bronchodilator properties could already be demonstrated several times and is accordingly not only an assumption of individual responders. Accordingly, the use in asthma is also found explicitly in your survey as one of the most frequently selected fields of application. https://doi.org/10.1111/pan.12577

„However, inhaled sedation should be avoided when there is a risk of increased intracranial pressure.” Considering that volatiles are used for the treatment of status epilepticus as well as in the neurosurgical ICU, I would be interesting to know the underlying recommendation. In your data, volatiles were used for the therapy of status epilepticus in 22% of the respondents. Please provide references.

I think it is problematic to primarily use the RAAS to monitor the level of sedation. This score may be helpful to measure shallow sedation or agitation, but deep sedation in severe ARDS in particular is not adequately captured by the assessment on which the score is based. Please specify your recommendations.

It might be worth mentioning that the titration of the desired sedation state using the minimum alveolar concentration with the closed loop system of the MIRUS system works without intervention depending on age, gender and weight, whereas this has to be done manually with the Anaconda device.

„always with a good satisfaction among the users” I think that it is not possible to speak of "always" with just under 74% of the users being very satisfied!

A problem not discussed appears to be the high number of ECMO treatments among COVID-19 patients and the impossible use of volatile sedatives in this patient population.

It should perhaps be mentioned that in the meantime activated charcoal systems exist on international markets to capture the volatile sedative no longer absorbed by the ACD, which even allows recycling. This would provide an approach to the ecological concerns expressed by 27 respondents.

6. PLOS authors have the option to publish the peer review history of their article (what does this mean?). If published, this will include your full peer review and any attached files.

Reviewer #1: No

Reviewer #2: No

---

## [Author Response · Author response to Decision Letter 0]

24 Oct 2022

RESPONSE TO EDITOR AND REVIEWER COMMENTS

Reference: MS#: PONE-D-22-06434

Title: Use of volatile anesthetics for sedation in the ICU during the COVID-19 pandemic: A national survey in France (VOL’ICU 2 study)

Raiko Blondonnet, Aissatou Balde, Ruoyang Zhai, Bruno Pereira, Emmanuel Futier, Jean-Etienne Bazin, Thomas Godet, Jean-Michel Constantin, Céline Lambert, Matthieu Jabaudon

We thank the Editors and the Reviewers for their careful reading and thoughtful comments on the previous manuscript version. We have carefully taken these comments into consideration in preparing our revision, and we hope the manuscript has been improved. Please find below a point-by-point response to the comments and questions.

JOURNAL REQUIREMENTS

C1. Please ensure that your manuscript meets PLOS ONE's style requirements, including those for file naming. The PLOS ONE style templates can be found at https://journals.plos.org/plosone/s/file?id=wjVg/PLOSOne_formatting_sample_main_body.pdf and https://journals.plos.org/plosone/s/file?id=ba62/PLOSOne_formatting_sample_title_authors_affiliations.pdf

R1. We have carefully proofread the manuscript in order to be in accordance with the PLOS ONE’s style requirements. 

C2. Thank you for stating the following in the Competing Interests section:

“The authors have read the journal’s policy and the authors of this manuscript have the following competing interests: MJ is a principal investigator of the SEvoflurane for Sedation in ARds (SESAR) (ClinicalTrials.gov Identifier: NCT04235608) and the ISCA study (ClinicalTrials.gov Identifier: NCT04383730), which are co-funded and funded, respectively, by grants from Sedana Medical. JMC and MJ received fees from Sedana Medical for participation in a scientific advisory panel; MJ received consulting fees from Abbvie. Other authors have no competing interest. There are no patents, products in development or marketed products to declare”

We note that one or more of the authors are employed by a commercial company: SESAR, Sedana Medical

a. Please provide an amended Funding Statement declaring this commercial affiliation, as well as a statement regarding the Role of Funders in your study. If the funding organization did not play a role in the study design, data collection and analysis, decision to publish, or preparation of the manuscript and only provided financial support in the form of authors' salaries and/or research materials, please review your statements relating to the author contributions, and ensure you have specifically and accurately indicated the role(s) that these authors had in your study. You can update author roles in the Author Contributions section of the online submission form. >>>

C2a. As requested, we have changed the Funding Statement and the Competing Interests Statement according to the journal requirements. Nevertheless, neither Matthieu Jabaudon (MJ) nor Jean-Michel Constantin (JMC) are employed by Sedana Medical or Abbvie. JMC and MJ received fees from Sedana Medical for participation in a scientific advisory panel; MJ received consulting fees from Abbvie.

C2b. As requested, we have changed the Funding Statement and the Competing Interests Statement according to the journal requirements. Neither Sedana Medical or Abbvie has no influence in the study and collection, analysis, and interpretation of data and in writing of the current study.

C3. Within your Competing Interests Statement, please confirm that this commercial affiliation does not alter your adherence to all PLOS ONE policies on sharing data and materials by including the following statement: "This does not alter our adherence to PLOS ONE policies on sharing data and materials.” (as detailed online in our guide for authors http://journals.plos.org/plosone/s/competing-interests) . If this adherence statement is not accurate and there are restrictions on sharing of data and/or materials, please state these. Please note that we cannot proceed with consideration of your article until this information has been declared.

R3. As requested, we included the recommended statement. 

R4. Please include both an updated Funding Statement and Competing Interests Statement in your cover letter. We will change the online submission form on your behalf.

C4. As requested, we included both an updated Funding Statement and Competing Interests Statement in your cover letter. 

EDITORIAL COMMENTS

REVIEWER COMMENTS

REVIEWER #1

C1. Title should be changed - from "...in the ICU AFTER COVID 19..." in "...in the ICU DURING COVID 19...." because the survey questions were related to COVID19 period.

R1. As suggested, we changed the title of the manuscript.

C2. Figure no. 1 has to changed into a more clear one. Legend should be chnaged (grey shades used can lead to mistakes - "no answer" and "> 100 patients" having a similar colour). Also the legend is not intelligible and not totally compatible with the text description

R2. We thank Reviewer #1 for this comment. As suggested, we updated both the Figure 1and the legend in order to improve the understanding. 

REVIEWER #2

Reviewer #2: I would like to express my gratitude for the opportunity to review the submitted intriguing work on the use of volatile sedation in France. Unfortunately, the work in its current form still has a number of shortcomings.

C3. I am sorry, but I have to criticise the inadequate referencing work. The study by your French colleague (reference 5 ) does not describe an actual shortage of sedatives! The colleagues describe aptly in their introduction to your study project: “Another major concern was to ensure access to mandatory anesthetics drugs medications, or neuromuscular blockade agents. In response to such shortages, we decided to diversify our sedative agents panel and thus to use volatile agents…”. However, authors should not be guided by the title of a paper without allowing its content to be presented in the referenced study! So far, the manuscript fails to provide appropriate references for the presentation of a real drug shortage. This applies equally to the corresponding section of the discussion.

R3. We apologize for the inadequate referencing work. We truly agree with Reviewer #2 about not to be guided only by the title of the paper and we applied the “reference 5 - doi: 10.1016/j.jcrc.2020.09.009” to the corresponding sentence. The “reference 5 - doi: 10.1016/j.jcrc.2020.09.009” argued that inhaled sedation could be an effective therapy in settings of medication shortages. As suggested, we added new references to report drug shortage worldwide in both the introduction and the discussion. 

C4. How do the authors conclude that volatile substances are an "abundant" resource? Whereas, according to them, intravenous sedatives are a scarce resource?

R4. We thank Reviewer #1 for this comment. Around 300 million surgical operations are performed globally worldwide each year and this number is increasing (https://doi.org/10.1016/j.ijsu.2020.07.017). Volatile anesthetics are used daily to provide general anesthesia in the operating room. Consequently, volatile anesthetics appear as an important actor in the hospital, which made us conclude that volatile anesthetics are an abundant resource, as suggested by Jerath et al. in their recent review (https://doi.org/10.1007/s00134-020-06154-8). As answered to the previous comment, the intravenous sedatives were a scarce resource during the shortage due to the COVID-19 pandemic. In order to avoid any confusion and to increase readability, we have rephrased the corresponding sentence.

C5. I'm not sure, but if I'm not mistaken, the MIRUS is the MIRUS device from the company Technologie Institut Medizin, which is distributed in by carelide, Dahlhausen, Anandic or Pall?

R5. We thank Reviewer #2 for this comment. Indeed, the MIRUS is the MIRUS device from the company Technologie Institut Medizin, which is currently distributed by Carelide (Mouvaux, France). Previously, the MIRUS device was distributed by Pall medical. 

C6. In another passage, the authors cite a well-done Narrative Review by the esteemed colleague Jerath on Volatile Sedation in COVID-19 (reference 11), which raises the hypothesis outlined, but does not substantiate it with appropriate COVID-19-specific data. Such data should present masterful performance 3 months after the global COVID-19 outbreak as well. A corresponding referencing with current data would be exciting.

R6. We thank Reviewer #2 for this comment. We added in the introduction some recent studies that have shown a potential beneficial effect of the inhaled sedation on the need for both intravenous sedatives and opioïds. Furthermore, we added a recent systematic review about inhaled sedation for invasively ventilated COVID-19 patients that lists the ongoing clinical trials on the effects of the inhaled sedation in ICU (https://doi.org/10.3390/jcm11092500).

C7. Regarding the acquisition of medical devices, it seems that 34% new acquisitions during the pandemic is a quite high rate of observation. However, it should be discussed that the purchase of a Sedaconda ACD membrane is around 16€ and a MIRUS device costs several thousand euros. Therefore, the MIRUS Device is not suitable for short-term acquisitions and would be an explanation for the clear dominance of the Anaesthetic conserving Device: AnaConDa/SeDaConDa.

R7. We thank Reviewer #2 for this interesting comment. As suggested, we added these elements to the Discussion.

C8. How do the authors explain that 47/62 respondents did not use the device due to lack of availability, when the majority of them used Sedaconda as a patient-specific single-use material?

R8. We agree with Reviewer #2 that this point is intriguing. It could explain that even if the industries have provided many devices all around France during the COVID-19 pandemic, some ICUs could experience shortages of some devices or consumables, thus restraining the use of inhaled sedation. We have rephrased the corresponding sentence in the Discussion.

C9. The authors disclose early on that numerous uses of volatile sedatives are under study conditions. Has it been evaluated on your part whether the center determining intravenous gas concentrations of volatiles is a study condition determination? I can't explain it any other way. And it does seem a bit confusing to someone not versed in the field.

R9. We agree with Reviewer #2 that is a difficult question and it could be confusing for “non-experts” in inhaled sedation, especially when implementing for the first time the use of inhaled sedation in the ICU. In this study, we did not assess the intravenous gas concentrations used via the expired fractions of sevoflurane declared used by the respondents under study condition. But, for example, in the SESAR study (https://doi.org/10.3390/jcm11102796), a large RCT conducted by our group, the level of sedation in the both groups (i.e., intravenous sedation or inhaled sedation) is protocolized using the Richmond Assessment Sedation Scale with the BIS value (if the patients are also the neuromuscular blockade). We rephrased the discussion in order to add this point. 

C10. “Thus, COVID-19 patients, especially when they develop ARDS, need higher doses of sedatives to reach the sedation objective compared to non-COVID19 patients [11].” As noted above, this source does not reflect the aggravated sedation per se, here the evidence has already been provided elsewhere: https://doi.org/10.1371/journal.pone.0253778/

R10. We thank Reviewer #2 for this comment and for the help. As suggested, we added this reference. 

C11. „The respondents also reported potential interest in the bronchodilator effect and…” also the bronchodilator properties could already be demonstrated several times and is accordingly not only an assumption of individual responders. Accordingly, the use in asthma is also found explicitly in your survey as one of the most frequently selected fields of application. https://doi.org/10.1111/pan.12577

R11. As suggested, we discussed the efficacy of inhaled sedation in patients with refractory life-threatening status asthmaticus and we added the reference. Thank you.

C12. However, inhaled sedation should be avoided when there is a risk of increased intracranial pressure.” Considering that volatiles are used for the treatment of status epilepticus as well as in the neurosurgical ICU, I would be interesting to know the underlying recommendation. In your data, volatiles were used for the therapy of status epilepticus in 22% of the respondents. Please provide references.

R12. We thank Reviewer #2 for this comment. As suggested, we added references about the use of volatile anesthetics for the treatment of refractory status epilepticus and for neurosurgical patients to the Discussion. 

C13. I think it is problematic to primarily use the RAAS to monitor the level of sedation. This score may be helpful to measure shallow sedation or agitation, but deep sedation in severe ARDS in particular is not adequately captured by the assessment on which the score is based. Please specify your recommendations.

R13. We thank the Reviewer #2 for this valuable comment and we agree that evaluation of the sedation is challenging in patients with ARDS. 

Before and after administering an NMBA, patients should receive an intravenous analgesic medication sufficient to provide acceptable pain relief, as well as a sedative to target a deep level of sedation. Sedation management in patients receiving NMBAs should ideally rely on validated scales or tools. However, the assessment of anxiety and pain when patients cannot communicate or express behavioral reactions is challenging. Among patients receiving NMBAs, neither the gold standard for pain assessment (i.e., the patient’s self-report) or recommended behavioral measures, such as the Behavioral Pain Scale (BPS) and the Critical-Care Pain Observation Tool (CPOT) can be used. Approaches for pain and sedation assessment in paralyzed patients are being explored such as the analgesia nociception index, or the pupillary pain index or the BIS-value. 

We rephrased this point in the discussion to improve the understanding. 

C14. It might be worth mentioning that the titration of the desired sedation state using the minimum alveolar concentration with the closed loop system of the MIRUS system works without intervention depending on age, gender and weight, whereas this has to be done manually with the Anaconda device.

R14. We thank the Reviewer #2 for this comment and as suggested we added this point in the introduction to improve the readability. 

C15. „always with a good satisfaction among the users” I think that it is not possible to speak of "always" with just under 74% of the users being very satisfied!

R15. We thank Reviewer #2 for this comment and as suggested we changed the word “always” by “frequently associated” in both the conclusion and the abstract. 

C16. A problem not discussed appears to be the high number of ECMO treatments among COVID-19 patients and the impossible use of volatile sedatives in this patient population.

R16. We thank Reviewer #2 for this interesting comment. Even if inhaled sedation in ARDS patients undergoing ECMO is relatively novel and raises several feasibility and safety questions, the use of volatile anesthetics in such a situation is possible and has ever been published in both COVID-19 and non COVID-19 patients. We added this important point in the discussion. 

C17. It should perhaps be mentioned that in the meantime activated charcoal systems exist on international markets to capture the volatile sedative no longer absorbed by the ACD, which even allows recycling. This would provide an approach to the ecological concerns expressed by 27 respondents.

R17. We thank Reviewer #2 for this important comment and we agree that environmental and occupational considerations of anesthesia and critical care medicine are very important. As suggested, we added in the discussion the need for a scavenging system, such as an activated carbon system.

---

## [Decision Letter · Decision Letter 1]

10 Nov 2022

Use of volatile anesthetics for sedation in the ICU during the COVID-19 pandemic: A national survey in France (VOL’ICU 2 study)

PONE-D-22-06434R1

Dear Dr. Blondonnet,

We’re pleased to inform you that your manuscript has been judged scientifically suitable for publication and will be formally accepted for publication once it meets all outstanding technical requirements.

Kind regards,

Silvia Fiorelli

Academic Editor

PLOS ONE

Additional Editor Comments (optional):

congratulations to the authors and thanks to the reviewers for the suggestions provided which really helped improve the quality of the manuscript

Reviewers' comments:

Reviewer's Responses to Questions

**Comments to the Author**

1. If the authors have adequately addressed your comments raised in a previous round of review and you feel that this manuscript is now acceptable for publication, you may indicate that here to bypass the “Comments to the Author” section, enter your conflict of interest statement in the “Confidential to Editor” section, and submit your "Accept" recommendation.

Reviewer #2: All comments have been addressed

2. Is the manuscript technically sound, and do the data support the conclusions?

Reviewer #2: Yes

3. Has the statistical analysis been performed appropriately and rigorously? 

Reviewer #2: Yes

4. Have the authors made all data underlying the findings in their manuscript fully available?

Reviewer #2: Yes

5. Is the manuscript presented in an intelligible fashion and written in standard English?

Reviewer #2: Yes

6. Review Comments to the Author

Reviewer #2: I once again would like to thank you for the opportunity to review this work.

Furthermore, I would like to thank the authors for their very conscientious revision.

It appears to be a matter of taste, but did the authors consider using colour instead of patterned greys for Figure1?

7. PLOS authors have the option to publish the peer review history of their article (what does this mean?). If published, this will include your full peer review and any attached files.

Reviewer #2: No

---

## [Editor Report · Acceptance letter]

18 Nov 2022

PONE-D-22-06434R1 

Use of volatile anesthetics for sedation in the ICU during the COVID-19 pandemic: A national survey in France (VOL’ICU 2 study) 

Dear Dr. Blondonnet:

I'm pleased to inform you that your manuscript has been deemed suitable for publication in PLOS ONE. Congratulations! Your manuscript is now with our production department. 

Kind regards, 

on behalf of

Dr. Silvia Fiorelli 

Academic Editor

PLOS ONE